# Distributions and Trends of the Global Burden of Colorectal Cancer Attributable to Dietary Risk Factors over the Past 30 Years

**DOI:** 10.3390/nu16010132

**Published:** 2023-12-30

**Authors:** Yuxing Liang, Nan Zhang, Miao Wang, Yixin Liu, Linlu Ma, Qian Wang, Qian Yang, Xiaoyan Liu, Fuling Zhou, Yongchang Wei

**Affiliations:** 1Department of Radiation and Medical Oncology, Zhongnan Hospital of Wuhan University, Wuhan 430071, China; liangyuxing@whu.edu.cn (Y.L.); wm94163200@163.com (M.W.); lyx19960626@163.com (Y.L.); yangqian@whu.edu.cn (Q.Y.); 2Department of Hematology, Zhongnan Hospital of Wuhan University, Wuhan 430071, China; zhang_nan@whu.edu.cn (N.Z.); malinlu702@whu.edu.cn (L.M.); wangqian@whu.edu.cn (Q.W.); liuxiaoyan@znhospital.cn (X.L.)

**Keywords:** global burden of disease, dietary risk factors, colorectal cancer, prevention

## Abstract

Dietary risk has always been a major risk factor for colorectal cancer (CRC). However, the contribution of dietary risk factors to CRC at the level of region, gender, and age has not been fully characterized. Based on the Global Burden of Disease (GBD) 2019 study, the death rates, age-standardized mortality rates (ASDRs), and estimated annual percentage changes (EAPCs) were calculated to assess the trends of CRC attributable to dietary risk factors over the past 30 years. Globally, the death cases of CRC increased to 1,085,797 in 2019, and the number of deaths attributed to dietary risk factors increased to 365,752 in 2019, representing approximately one-third of all CRC-related fatalities. Overall, the ASDR attributable to dietary risks was 4.61 per 100,000 in 2019, with a slight downward trend (EAPC = −0.29). Notably, there is a rising trend in early-onset colorectal cancer mortality associated with dietary factors. To alleviate CRC burdens, it is recommended to elevate the intake of whole grains, milk, calcium, and fiber while reducing consumption of red and processed meats. The results will improve the understanding, and provide guidance on the diet of CRC in different regions, gender, and age groups worldwide.

## 1. Introduction

Globally, colorectal cancer (CRC) ranks third in incidence and second in mortality, with nearly 1.9 million new cases and 935,000 deaths in 2020 [1]. In general, the number of CRC-related incident cases and death cases has increased globally. The incidence and mortality rates have mostly declined in high SDI countries, while age-standardized rates have increased in some low SDI and low-middle SDI countries and regions [2,3,4]. Differences in the incidence and mortality rates of colorectal cancer (CRC) worldwide correlate with various levels of human development, revealing widening disparities and escalating burdens, particularly in transitioning countries [5].

Studies have shown that the occurrence of CRC is related to a variety of factors, such as unhealthy diets, alcohol consumption, smoking, physical inactivity, etc. [3,6,7]. Unhealthy food or so called diet type is one of the main risks contributing to the increased prevalence of CRC worldwide in recent years, especially among younger populations, where unhealthy dietary patterns may increase the risk of CRC [8,9,10,11,12,13]. For example, high intake of simple sugars and sugar-sweetened beverages during adolescence was found to be associated with an increased risk of rectal adenoma [14]. Several studies have confirmed that dairy products and calcium are inversely associated with CRC risk [12]. In a case-control study, adherence to a Mediterranean diet (a diet with low intake of sugar-sweetened beverages and red meat and high intake of fish) was associated with lower odds of advanced colorectal polyps [15]. However, a more precise and comprehensive comprehension of the dietary risks linked to CRC remains incomplete, underscoring the indispensable nature of relying on epidemiological studies to shape dietary recommendations and guide medical decision-making.

The Global Burden of Disease (GBD) Study was initiated in the 1990s for the purpose of assessing health outcomes in a timely, valid, and relevant manner [16,17,18,19]. There are thousands of outcomes assessed in the latest version of GBD pertaining to diseases, injuries, and risk factors in 204 countries and territories [20]. In this study, epidemiological data concerning CRC based on gender, region, country, age, and dietary risks were extracted from the GBD. Understanding the disease burden of CRC attributable to the dietary risks is important for governments and health organizations. It helps in developing CRC prevention strategies and guiding individuals to adopt healthy dietary habits.

## 2. Materials and Methods

### 2.1. Data Sources

GBD 2019 represents a multinational collaborative integrated surveillance system, providing comprehensive and systematic estimates of 369 diseases and injuries as well as 87 risk factors from 1990 to 2019 [21,22]. For this study, we acquired estimates of deaths attributed to dietary risk factors based on SDI, age, location, and sex in CRC from the Global Health Data Exchange query tool (http://ghdx.healthdata.org/gbd-results-tool, accessed on 22 January 2022 ) [20]. The SDI serves as a metric gauging per capita income, educational attainment, and total fertility rate in each state, with a higher index signifying improved socio-demographic development.

### 2.2. Dietary Risk Factor Definitions

The GBD dietary data were based on the FAO Food Balance Sheet (FBS) and Global Nutrient Database, In GBD 2019, dietary risk factors were updated using new sources like PubMed and IHME GHDx surveys. Data sources included national surveys, household budgets, Euromonitor sales, and UN FAO Supply and Utilization Accounts [23]. Through systematic reviews of nationally representative nutrition surveys, scientific literature, and household budget surveys, GBD collected the consumption due to every dietary factor around the world [19]. Among the 15 dietary risk factors outlined in the GBD comparative risk assessment (CRA) framework, 6 risk factors were associated with CRC outcomes. These include diets low in whole grains, milk, fiber, and calcium, as well as diets high in red and processed meats [4]. Among the gold standard data sources for these dietary risks are 24 h dietary recall surveys where food and nutrient intake are documented or converted to grams per person per day. Appendix A outlines the theoretical minimum risk exposure level for each dietary risk factor.

### 2.3. Statistical Analysis

The count of deaths, age-standardized death rates (ASDRs), and the estimated annual percentage changes (EAPCs) were used to quantify the global burden of CRC attributable to dietary risk factors [3,24,25]. ASDR represents the death rate per 100,000 people, using a standardized global age structure that eliminates the effects of changes in age structure [26]. EAPC is widely accepted to reflect annual interest rate changes over a specific period, and it is calculated based on a linear regression model [25]. An upward trend in the ASDR is evidenced if both the EAPC estimate and the 95% confidence intervals (CIs) are positive and vice versa [16]. The formulae of ADSR and EAPC are described below, where *α_k_* refers to the distribution of the selected reference standard population in the *k* age group, *β_k_* refers to the age-specific rate, *y* = *ln* [ASDR], *x* is the calendar year, *ε* is the error term, and *b* represents the positive or negative trend of ASDR in the linear formulation:$$\mathrm{ASDR}=\frac{\sum_{k=1}^{n}\alpha_{k\;}\beta_{k}}{\sum_{k=1}^{n}\beta_{k}}\times 100,000$$
y=a+bx+ε
$$EAPC=100\%\times (e^{b}-1)$$

For this study, all data analyses and visualization were performed by the software of GraphPad Prism (version 8.02) and the R software (version 3.5.2).

## 3. Results

### 3.1. Global Burden of Colorectal Cancer

Globally, the total death cases due to CRC increased steadily from 518,126 in 1990 to 1,085,797 in 2019, but the ASDR of CRC showed a slight downward trend (EAPC = −0.21) (Table 1 and Figure 1A). Meanwhile, with the gradual improvement of SDI, there was an observed increase in deaths associated with CRC. However, a notable shift occurred in 2019, wherein the high SDI region experienced a downward trend in this regard (Table 1 and Figure 1B). Moreover, ASDR for deaths of CRC was stable from 1990 to 2019, similarly the ASDR was on the rise in the low-middle SDI region (EAPC = 1.15) and middle SDI region (EAPC = 1.24) but declined in the high-SDI region (EAPC = −1.09) (Table 1 and Figure 1C). Therefore, CRC-related ASDRs increased in 15 GBD regions, with the largest increase being observed in East Asia (EAPC = 1.4), followed by Southeast Asia (EAPC = 1.25) and Andean Latin America (EAPC = 1.16). Central Europe had the highest ASDR (23.57 per 100,000) in the world in 2019. In contrast, declines in ASDR were observed in Australasia (EAPC = −1.73), high-income North America (EAPC= −1.22) and Western Europe (EAPC = −1.09), South Asia had the lowest ASDR (7.29 per 100,000) at the end of this study period (Table 1). At the country level, the highest ASDRs were observed in Greenland (31.38 per 100,000), Brunei Darussalam (30.26 per 100,000), and Hungary (28.56 per 100,000). On the other hand, the lowest ASDRs were found in Bangladesh (4.94 per 100,000), Somalia (5.01 per 100,000), and Nepal (5.4 per 100,000) in 2019 (Figure 1D and Appendix A). Significantly, the CRC-related ASDRs saw an increase in over half of the world, with the most substantial rises noted in Equatorial Guinea (EAPC = 3.51), Viet Nam (EAPC = 2.64), and Cabo Verde (EAPC = 2.64). Meanwhile, ASDRs in several countries, including Equatorial Guinea, Viet Nam, and Bulgaria, were both high and had increased rapidly over 30 years which may indicate major healthcare challenges in these countries. In contrast, several countries such as Austria, Singapore, and Luxembourg, exhibited the fastest ASDR declines, with EAPCs as low as −1.99 to −2.81 (Figure 1E and Appendix A).

In general, a greater burden of CRC was observed in males than in females, and the sex disparity may increase as time goes on. Both men and women experienced a global increase in total deaths in 2019, with male deaths reaching 594,176 and female deaths reaching 491,622 (Table 1). In addition, the death cases in males were consistently higher than females, with the gap increasing over time (Figure 1A). Meanwhile, the ASDRs in males were higher than in females at the regional levels. The highest ASDRs were observed in Central Europe (32.4 per 100,000) in males and Southern Latin America (17.5 per 100,000) in females, while the lowest ASDRs were observed in South Asia (7.3 per 100,000) in males and Central Sub-Saharan Africa (6.4 per 100,000) in females (Figure 2A). As shown in Figure 2B,D, the ASDR in males was observed to increase from 1990 to 2019 globally except to decline significantly in the high SDI region. Nevertheless, there was a global decrease in ASDR among females, particularly in the high-middle and high SDI regions (Figure 2C,D). Moreover, between 1990 and 2019, the male ASDR to female ASDR ratio was essentially maintained between 1.00 and 1.20 in low and low-middle SDI regions. However, from middle to high SDI regions, the ratios were higher than 1.40 in 2019 (Figure 2E), further indicating the gender-based death imbalance burden of CRC in developed regions.

### 3.2. Overall Impact of Dietary Risk Factors on Colorectal Cancer Burden

In recent years, mounting evidence has underscored the significant impact of diet on both the incidence and prevention of CRC [4,11,27,28]. Globally, the total death cases of CRC attributed to all dietary risks increased steadily from 179,639 in 1990 to 365,752 in 2019, but the ASDR showed a slight downward trend (EAPC = −0.29) (Appendix A). Meanwhile, the ASDR of deaths attributable to all dietary risks increased alongside the gradual enhancement of SDI over the past 30 years, however, this trajectory shifted with a downward trend observed in the high SDI region in 2019. Similarly, the ASDR is on the rise in the low SDI region (EAPC = 0.44), low-middle SDI region (EAPC = 0.89), and middle region (EAPC = 0.96) but declined in the high-middle region (EAPC = −0.32) and high-SDI region (EAPC = −1.04). Furthermore, in 21 GBD regions, the largest increase in ASDR was observed in East Asia (EAPC = 1.15) and the largest decline was observed in Australasia (EAPC = −1.66) (Appendix A).

At the country level, the highest ASDRs of deaths attributable to dietary risk factors were observed in Greenland (9.78 per 100,000), Brunei Darussalam (9.54 per 100,000), and Hungary (9.35 per 100,000), which is consistent with the ASDRs of CRC as described earlier. On the contrary, the lowest ASDRs were found in Nepal (1.99 per 100,000), Niger (2.01 per 100,000), and Bangladesh (2.11 per 100,000) in 2019 (Figure 3A and Appendix A). Meanwhile, the largest ASDR of dietary risk factor increases were observed in Equatorial Guinea (EAPC = 2.62), Lesotho (EAPC = 2.34), and Viet Nam (EAPC = 2.33). In contrast, several countries such as Austria (EAPC = −2.93), Singapore (EAPC = −2.25), and Luxembourg (EAPC = −2.06) showcased the most rapid declines in ASDR related to dietary risks, which were the same as the ASDRs in all risks (Figure 3B and Appendix A). Furthermore, there was a significant positive linear association between ASDR of colorectal cancer attributable to dietary risk and SDI by regions (ρ = 0.757, *p* < 0.001) and by nations (ρ = 0.56, *p* < 0.001) (Figure 3C,D). The proportion of dietary risk factors did not notably differ among the three age groups. However, it decreased from 1990 to 2019, with the most substantial decline noted in the 75+ age group within the middle SDI region. In 21 GBD regions in Eastern Europe, there was observed a significant decline trend across all age groups. In East Asia, a decline was observed in the 50–74 and 75+ year group (Figure 3E). Notably, significant increases in mortality were observed in males in the 15–49 year group in middle SDI and high-middle SDI regions. This highlights the need for heightened attention among this demographic and underscores the importance of adopting preventive dietary measures (Figure 3F).

The top three risk factors contributing to ASDRs across all SDI regions are diets which are low in whole grains, milk, and calcium (Figure 4A). These three risk factors slowly increased from 1990 to 2019 in low, low-middle, and middle SDI regions, with EAPCs from 0.3 to 1.24. Simultaneously, ASDRs attributed to diets low in whole grains and calcium remained persistently high, yet experienced rapid declines in the high-middle and high SDI regions, with EAPCs ranging from −0.34 to −0.97. However, in the high-middle SDI region, ASDR, attributed to diet low in milk, showed a continuously increasing trend (EAPC = 0.55), and conversely, there was a downward trend in the high SDI region (EAPC = −1.14). Diet high in red meat was ranked fourth among the six diet risk contributors to CRC-related deaths in 2019. During the past decades, diet high in red meat attributed to death rates increased in the low, low-middle, and middle SDI regions but decreased significantly in the high SDI region. Thus, the global trend of the ASDR of diet high in red meat tended to remain flat (Figure 4A and Table 2). Globally, the gender ratio (male to female) has shown a significant upward trend in the past 30 years, which could be attributed to all the six diet risks. The sex ratio for a diet low in calcium (male to female > 1.5) significantly surpassed that of other risk factors, signifying that a diet low in calcium notably contributed to the sex-dependent variations in deaths attributed to CRC (Figure 4B). At the global level, most of the deaths attributed to the six dietary risk factors showed an increasing trend, especially in the low-middle and middle SDI regions. Notably, in the low SDI region, the increase in deaths in the 15–49 year group was greater than that in the 50–74 year group, indicating that young people in the low SDI region need to pay more attention to a healthy diet (Figure 4C).

### 3.3. Colorectal Cancer Burden Attributed to Each Dietary Risk Factor

#### 3.3.1. Diet Low in Whole Grains

Globally, the death cases of CRC attributed to diet low in whole grains increased steadily from 83,524 in 1990 to 171,487 in 2019, but the ASDR showed a slight downward trend (EAPC = −0.29) (Appendix A). Similarly, the ASDR is on the rise in the low SDI region (EAPC = 0.56), low-middle SDI region (EAPC = 1.08), and middle SDI region (EAPC = 1.24) but declined in the high-middle SDI region (EAPC = −0.34) and high SDI region (EAPC = −0.97) (Appendix A). At the country level, the highest ASDRs attributable to diet low in whole grains were observed in Greenland (4.83 per 100,000), Hungary (4.76 per 100,000), and Uruguay (4.72 per 100,000) (Appendix A). Meanwhile, the largest ASDRs of CRC attributable to diet low in whole grains increases were observed in Equatorial Guinea (EAPC = 3.05), Cabo Verde (EAPC = 2.56) and Viet Nam (EAPC = 2.55) (Appendix A). Furthermore, the ASDR attributable to diet low in whole grains was significantly positive linearly correlated with the SDI by regions (ρ = 0.774, *p* < 0.001) and by nations (ρ = 0.693, *p* < 0.001) (Appendix A). The percentage of deaths related to a diet low in whole grains showed no significant differences among the three age groups (Appendix A). Notably, significant increases in mortality were observed in males in the 15–49 year group in the middle SDI and high-middle SDI regions (Appendix A).

#### 3.3.2. Diet Low in Milk

The death cases of CRC attributed to diet low in milk increased steadily from 72,199 in 1990 to 166,456 in 2019, and the ASDR showed a slight upward trend (EAPC = 0.19) over the past 30 years (Appendix A). Meanwhile, the ASDR attributable to diet low in milk increased in the low SDI region (EAPC = 0.46), low-middle SDI region (EAPC = 1.13), middle SDI region (EAPC = 1.13), and high-middle SDI (EAPC = 0.55) but declined in the high-middle SDI region (EAPC = −1.14) (Appendix A). At the country level, the highest ASDRs attributable to diet low in milk were observed in Greenland (4.83 per 100,000), Hungary (4.76 per 100,000), and Uruguay (4.72 per 100,000) (Appendix A). The largest ASDRs of deaths attributable to diet low in milk increases were observed in Equatorial Guinea (EAPC = 3.05), Cabo Verde (EAPC = 2.56) and Viet Nam (EAPC = 2.55) (Appendix A). Similarly, there was a positive linear association between ASDR of CRC by regions (ρ = 0.353, *p* < 0.001) and by nations (ρ = 0.253, *p* < 0.001) (Appendix A). In the 21 GBD regions, the percentage is highest in Central Sub-Saharan Africa (Appendix A). Furthermore, significant increases in mortality were observed in all age groups in low to high-middle SDI regions; especially in the male group, the increasing trend was more significant (Appendix A).

#### 3.3.3. Diet Low in Fiber

The number of CRC-related deaths attributed to diet low in fiber increased from 12,548 in 1990 to 20,499 in 2019. However, the ASDR displayed a notable downward trend during this period (EAPC = −1.13) (Appendix A). Meanwhile, the ASDR attributable to diet low in milk decreased in the low-middle SDI region (EAPC = −0.28), middle SDI region (EAPC = −0.61), high-middle SDI (EAPC = −0.92), and high SDI region (EAPC = −1.5) but increased in the low SDI region (EAPC = 0.12) (Appendix A). At the country level, ASDRs attributed to diet low in fiber were highest in Saint Kitts and Nevis (1.21 per 100,000), Viet Nam (1.16 per 100,000), and Cambodia (1.13 per 100,000) (Appendix A). The largest ASDRs of deaths attributable to diet low in fiber increases were observed in Democratic Republic of the Congo (EAPC = 3.55), Iraq (EAPC = 3.13) and Ecuador (EAPC = 2.9) (Appendix A). Similarly, the ASDR of CRC attributable to diet low in fiber had a mildly positive linear correlation with the SDI by regions (ρ = 0.592, *p* < 0.001) and by nations (ρ = 0.309, *p* < 0.001) (Appendix A). In the 21 GBD regions, the death percentage is highest in Southeast Asia for all age groups (Appendix A). Furthermore, the death rates were decreasing almost in all age groups, but significant increases in mortality were observed in the 15–49 year group in the middle SDI region, especially in males, suggesting the need to increase fiber intake in this population (Appendix A).

#### 3.3.4. Diet Low in Calcium

Globally, the death cases of CRC attributed to diet low in calcium increased from 63,709 in 1990 to 137,896 in 2019, and the changes of the ASDR remained relatively stable (EAPC = 0.04) from 1990 to 2019 (Appendix A). The ASDR attributable to diet low in calcium increased in the low SDI region (EAPC = 0.3), low-middle SDI region (EAPC = 0.65), and middle SDI region (EAPC = 0.51) but decreased in the high-middle SDI (EAPC = −0.37) and high SDI region (EAPC = −0.85) (Appendix A). At the country level, the highest ASDRs attributable to diet low in fiber were observed in Saint Kitts and Nevis (1.21 per 100,000), Viet Nam (1.16 per 100,000) and Cambodia (1.13 per 100,000) (Appendix A). The largest ASDR of deaths attributable to diet low in fiber increases were observed in Democratic Republic of the Congo (EAPC = 3.55), Iraq (EAPC = 3.13) and Ecuador (EAPC = 2.9) (Appendix A). Similarly, the ASDR of CRC attributable to diet low in fiber had a mildly negative linear correlation with the SDI by nations (ρ = −0.275, *p* < 0.001) but had no linear correlation by regions (ρ = −0.045, *p* = 0.236) (Appendix A). The death percentage of diet low in fiber showed little difference between the three age groups, but had an increasing trend in the 50–74 and 75+ year group from 1990 to 2019 (Appendix A). Furthermore, the global death rates were decreasing in the 50–74 and 75+ year age groups, but significant increases in mortality were observed in the 15–49 year group especially in males (Appendix A).

#### 3.3.5. Diet High in Red Meat

Between 1990 and 2019, cases of CRC-related deaths due to a high red meat diet surged from 26,087 to 52,811. However, the ASDR exhibited a slight downward trend (EAPC = −0.32) over the same period (Appendix A). The ASDR attributed to diet high in red meat increased in the low SDI region (EAPC = 0.81), low-middle SDI region (EAPC = 2.38) and middle SDI region (EAPC = 2.95) but decreased in the high SDI region (EAPC = −1.29), with being relatively stable in the high-middle SDI region (EAPC = 0.09). At the country level, the highest ASDRs attributed by diet high in red meat were observed in Argentina (2.61 per 100,000), Monaco (2.46 per 100,000) and Greenland (2.36 per 100,000) (Appendix A). The largest ASDR of deaths attributed by diet low in fiber increases were observed in Equatorial Guinea (EAPC = 4.71), Viet Nam (EAPC = 4.5) and Myanmar (EAPC = 4.29) (Appendix A). Similarly, the ASDR of CRC attributed to diet high in red meat had a significantly positive linear correlation with the SDI by regions (ρ = 0.679, *p* < 0.001) and by nations (ρ = 0.623, *p* < 0.001) (Appendix A). The death percentage of diet low in fiber showed little difference between the three age groups but had an increasing trend in the 15–49 year group in 2019. In the 21 GBD regions, the percentage is highest in Australasia and lowest in South Asia for all age groups (Appendix A). Furthermore, the global death rates attributable to diet high in red meat were increasing in the 15–49 year group, especially in males and in the low to high middle regions (Appendix A).

#### 3.3.6. Diet High in Processed Meat

For diet high in processed meat, the death cases of CRC increased from 20,185 in 1990 to 33,928 in 2019, and the changes in the ASDR remained a significant downward trend (EAPC = −1.08) from 1990 to 2019 (Appendix A). The ASDR attributed to diet high in processed meat increased in the low SDI region (EAPC = 0.89), low-middle SDI region (EAPC = 1.79), middle SDI region (EAPC = 2.47) but decreased in the high-middle SDI region (EAPC = −0.88) and high SDI region (EAPC = −1.04) (Appendix A). At the country level, the highest ASDRs attributed to diet high in processed meat were observed in Greenland (2.08 per 100,000), Monaco (1.56 per 100,000), and Norway (1.55 per 100,000) (Appendix A). The largest ASDRs of deaths attributed to diet low in fiber increases were observed in Equatorial Guinea (EAPC = 5.0), Viet Nam (EAPC = 3.7), and Romania (EAPC = 3.56) (Appendix A). Similarly, the ASDR of CRC attributed by diet high in processed meat exhibited a notably positive linear correlation with the SDI by regions (ρ = 0.785, *p* < 0.001) and by nations (ρ = 0.602, *p* < 0.001) (Appendix A). The death percentage of diet low in fiber showed little difference between the three age groups with a downward trend from 1990 to 2019, and the largest percentage was observed in the high SDI region. In the 21 GBD regions, the percentage is highest in high-income North America for all age groups (Appendix A). Furthermore, the global death rates attributed to diet high in processed meat were decreasing almost in all age groups (Appendix A).

## 4. Discussion

As tumors pose a significant threat to global human health, particularly CRC as a gastrointestinal malignancy, its worldwide burden continues to increase [29,30]. While previous articles have elucidated the colorectal cancer burden over the past three decades [3], our study specifically focuses on dissecting the impact of dietary factors on global colorectal cancer incidence and trends. By exploring intricate connections between different dietary elements and the prevalence of this malignancy, our research offers detailed insights. In this study, we used the GBD database to evaluate the disease trends and burden of CRC deaths attributed to dietary risks. Our study showed that there were 365,752 deaths of CRC attributed to dietary risk in 2019, accounting for about one-third of all CRC deaths. Geographically, there were resemblances between the distributions depicted in Figure 1D and Figure 3A, suggesting a strong association between dietary risk factors and CRC-related fatalities. This study selected all dietary risk factors associated with CRC from GBD 2019, including diet low in whole grains, diet low in milk, diet low in fiber, diet low in calcium, diet high in red meat, and diet high in processed meat. In line with our results, an umbrella review of meta-analyses showed that red meat consumption was positively associated with the incidence of CRC, while higher intakes of dietary fiber, calcium, and yogurt were inversely associated with the incidence of CRC [24]. The World Cancer Research Fund (WCRF) has emphasized that red and processed meat are associated with the development of CRC and that any safe level of processed meat cannot be confidently attributed to lack of risk [31,32].

Diet remains a pivotal environmental factor closely associated with colorectal cancer (CRC), yet there has been a limited exploration of gender-specific differences in dietary influences on CRC progression [33]. Our study’s insights uncovered a higher mortality linked to all dietary risks among men compared to women across diverse age groups. Delving into the six dietary factors, the gender disparity persisted notably, especially evident in diets low in milk and calcium, where the variance was more pronounced. The precise determinants behind this gender divergence remain enigmatic, with genetic and environmental factors emerging as crucial contributors [33,34]. Men often develop left-sided colon tumors, while women tend to exhibit right-sided tumors. Recent investigations have shed light on the prevalence of right-sided (proximal) colon cancer being more prevalent in females, contrasting the higher incidence of left-sided (distal) colon cancer in males [35,36,37]. Existing research linking escalated meat consumption to a heightened risk of left-sided colon cancer corroborates our study’s findings [38,39]. Furthermore, variances in dietary habits, especially higher red and processed meat intake among men, contribute to increased CRC risk. Additionally, greater calcium intake, prevalent among women, correlates with lowered CRC risk. Considerable evidence indicates a reverse link between increased calcium consumption and distal colon cancer risk [40,41,42], highlighting escalated mortality linked to inadequate calcium intake in men compared to women. Biological dissimilarities, including hormonal influences, also play a role in CRC development. These complex factors intertwine, impacting CRC incidence and mortality disparately between genders. However, further in-depth, gender-specific inquiries into the intricate interplay of dietary risks are imperative to furnish comprehensive guidelines for CRC prevention. The intricate relationship between dietary factors and gender disparity in CRC mortality underscores the necessity for targeted investigations to refine dietary recommendations and interventions tailored to individualized risk profiles.

Interestingly, in this study, we observed that the global CRC mortality rate attributable to dietary risk factors showed an overall downward trend (EAPC = −0.29), but the mortality rate in the 15–49 year group showed a significant upward trend, in which the increasing trend was more pronounced in men than in women (Figure 3F). Meanwhile, although the global incidence of CRC is declining, the incidence in young people has been increasing since the 1990s [8,43,44]. The rise in early-onset colorectal cancer mortality could be attributed to diverse factors including unhealthy lifestyle choices, genetic predispositions, and environmental exposures. Specific dietary habits, such as increased intake of red and processed meats along with inadequate fiber consumption, may contribute to this trend. Moreover, insufficient screening and delayed diagnosis in younger individuals might exacerbate mortality rates related to early-onset colorectal cancer. Obesity, and lack of exercise were probable major contributors to CRC in young adults [45,46]. Moreover, in the 15–49 age group, increasing trends in mortality attributable to diet were observed in almost all regions except the high SDI region. In the low SDI region, however, the 15–49 year group experienced a larger percentage increase in deaths than the 50–74 year group (Figure 4C). We speculate that in countries or regions with lower economic levels, inappropriate dietary patterns and low adherence to specific CRC screening programs among young adults may be major contributors [44]. Our findings suggest that young adults, especially males, need to promote the consumption of whole grains and fiber, limit the consumption of red and processed meat, and adopt a more balanced dietary pattern.

The ASDR for colorectal cancer exhibits a declining trend in high Socio-demographic Index (SDI) regions while showing an upward trajectory in other SDI areas. This disparity primarily results from socio-economic imbalances, inadequate healthcare investment, and poor health awareness exacerbating the burden in low SDI regions [47]. Addressing this divergence necessitates global cooperation and partnerships to improve healthcare infrastructure and promote awareness in less developed regions.

The accuracy of colorectal cancer models often relies on the quality of input data [48,49,50,51]. Although findings from the Global Burden of Disease (GBD) Study have been utilized in retrospective trials, the World Health Organization (WHO) does not universally endorse all aspects of the GBD Study due to certain disputes or contentions regarding its conclusions or methodologies. Data published by GLOBOCAN indicates over 1.9 million new cases of colorectal cancer and an estimated 930,000 deaths in 2020, and the highest mortality was observed in Eastern Europe (20.2 per 100,000, males) and the lowest in Southern Asia (2.5 per 100,000, females). This closely aligns with GBD data and maintains consistency in geographical distribution [29]. This underscores the importance of data accuracy and highlights the observed similarities between GLOBOCAN and GBD data despite certain methodological controversies. It should be noted that dietary risk factors are not considered in the current modeling strategies for CRC-related deaths in GBD. Thus, our analysis relied on all-cause death attributed to CRC dietary risk factors rather than CRC-related deaths. In addition, whole grains are rich in fiber, so there exists overlap with the fiber group. The wide variety of whole grain products makes accurate measurement of intake challenging, which can lead to measurement errors. Despite these limitations, we believe that GBD overcomes these limitations and provides reasonable estimates.

## 5. Conclusions

In summary, CRC remains a significant global public health concern. While the ASDR for CRC has been decreasing worldwide, the rise in age-standardized incidence across most nations presents a substantial public health challenge on a global scale. Our study found that even though the ASDR of CRC attributable to dietary risks was decreasing globally, there was still a clear upward trend in CRC deaths attributable to dietary factors in low, low-middle, and middle SDI regions. Furthermore, the burden of CRC deaths attributed to dietary risk was generally higher in males, and the gender difference became more pronounced over time. In addition, the disease burden of CRC attributable to dietary risk showed an increasing trend in the young age group, especially in males. This study provides a comprehensive assessment of CRC trends due to dietary risk factors over the past three decades, and the results have provided the basis for future research in this field and may have a positive impact on the development of subsequent prevention and treatment strategies, especially in countries with a high incidence of increased burden.

## Figures and Tables

**Figure 1 nutrients-16-00132-f001:**
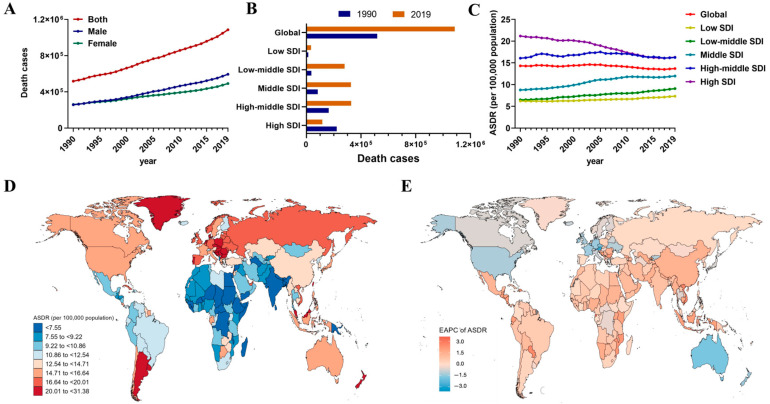
The death trends and gender differences of CRC in different regions from 1990 to 2019. (**A**) Death cases of CRC by sex. (**B**) Deaths in territories with low to high SDIs in 1990 and 2019. (**C**) The ASDR in different SDI regions from 1990 to 2019. (**D**) The ASDR in 204 countries and territories in 2019. (**E**) The EAPC in ASDR in 204 countries and territories from 1990 to 2019.

**Figure 2 nutrients-16-00132-f002:**
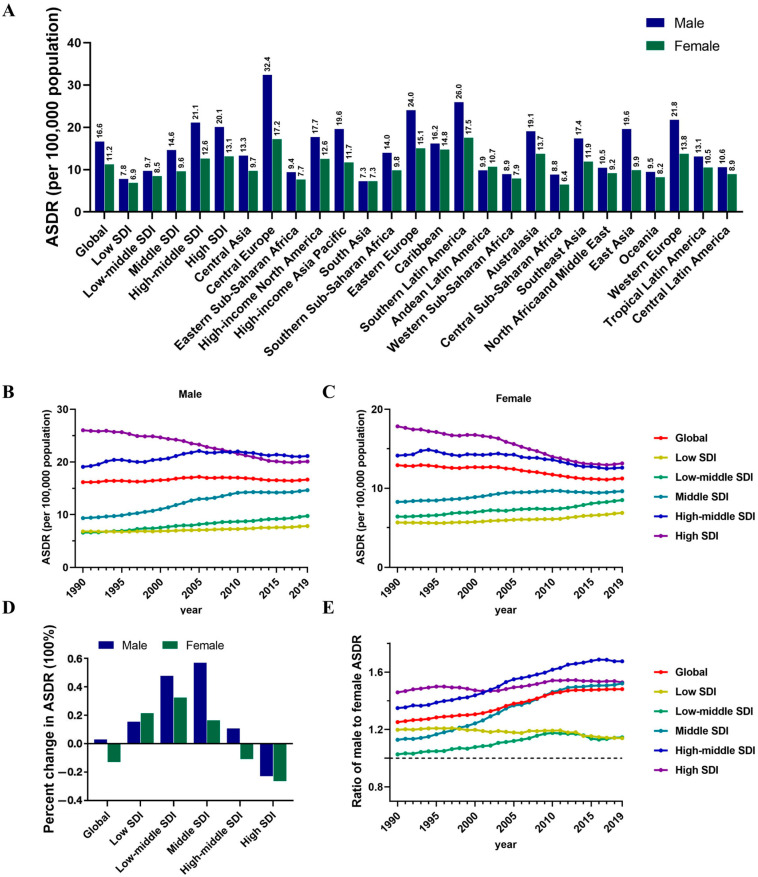
Sex differences and trends in CRC death rates in different regions. (**A**) The ASDR in males and females globally, in territories with low to high SDIs and in 21 GBD regions in 2019. (**B**) The ASDR globally and in territories with low to high SDIs for males from 1990 to 2019. (**C**) The ASDR globally and in territories with low to high SDIs for females from 1990 to 2019. (**D**) The percentage changes in ASDR in males and females between 1990 and 2019. (**E**) Male to female ratios of ASDR globally and in territories with low to high SDIs from 1990 to 2019.

**Figure 3 nutrients-16-00132-f003:**
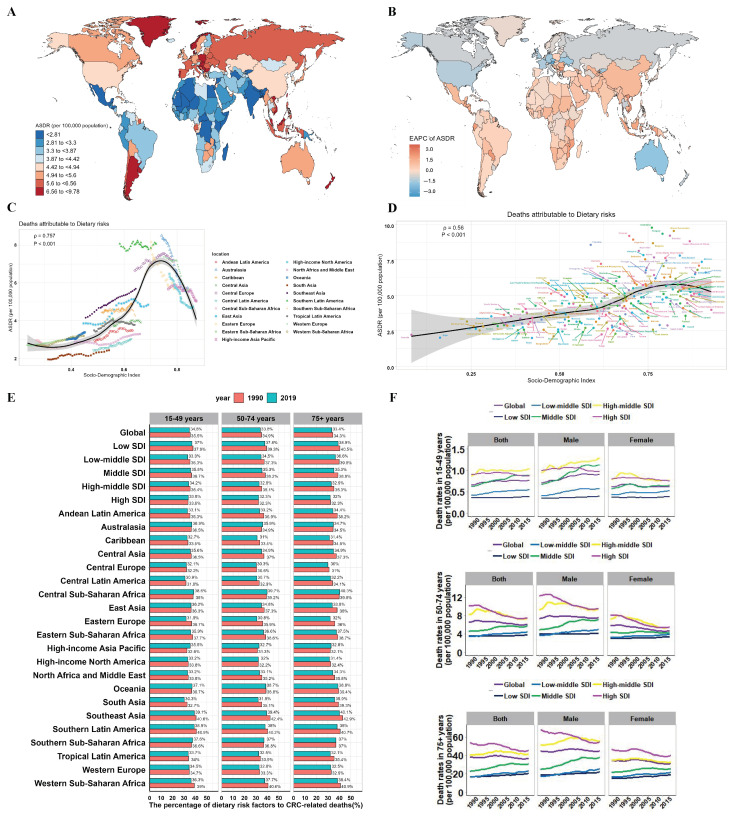
The global burden of CRC attributable to dietary risk over the past 30 years. (**A**) The ASDR in 204 countries and territories in 2019. (**B**) The EAPC in ASDR in 204 countries and territories from 1990 to 2019. (**C**) Relationship between SDI and ASDR of CRC attributable to dietary risk by regions in 2019. (**D**) Relationship between SDI and ASDR of CRC attributable to dietary risk by nations in 2019. (**E**) The percentage of dietary risk factors to CRC-related deaths between 1990 and 2019, in territories with low to high SDIs and in 21 GBD regions. (**F**) Deaths attributable to dietary risk in CRC by different age and sex from 1990 to 2019.

**Figure 4 nutrients-16-00132-f004:**
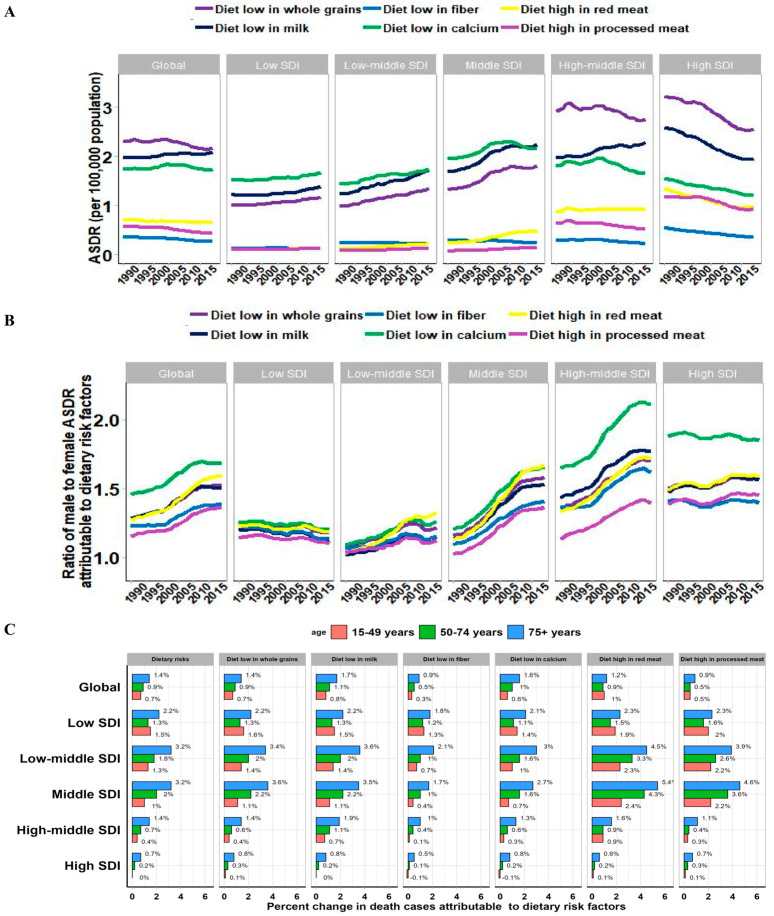
Predominant contribution of dietary risk factors to CRC-related deaths by SDI, sex, and age groups. (**A**) The ASDR attributable to main dietary risk factors by SDI region in 2019. (**B**) Male to female ratios of ASDR attributable to dietary risk factors in 2019. (**C**) The percent changes in deaths attributable to dietary risk factors by age group and SDI region between 1990 and 2019.

**Table 1 nutrients-16-00132-t001:** The deaths and ASDR of CRC in 1990 and 2019, and their temporal trends.

Characteristics	1990		2019		1990–2019
	Number of Deaths (95% UI)	ASDR per 100,000 (95% UI)	Number of Deaths (95% UI)	ASDR per 100,000 (95% UI)	EAPC of ASDR (95% CI)
Global	518,126 (493,682 to 537,877)	14.31 (13.52 to 14.88)	1,085,797 (1,002,795 to 1,149,679)	13.69 (12.6 to 14.51)	−0.21 (−0.28 to −0.14)
Sex					
Male	257,310 (246,262 to 271,018)	16.16 (15.44 to 16.93)	594,176 (550,959 to 638,031)	16.64 (15.39 to 17.85)	0.1 (0.03 to 0.17)
Female	260,816 (244,563 to 275,303)	12.91 (12.03 to 13.64)	491,622 (437,555 to 532,378)	11.24 (10.01 to 12.17)	−0.59 (−0.67 to −0.52)
Socio-demographic index					
Low SDI	1,3443 (11,307 to 15,790)	6.22 (5.21 to 7.26)	34,656 (30,959 to 38,606)	7.33 (6.53 to 8.14)	0.58 (0.52 to 0.64)
Low-middle SDI	36,051 (32,664 to 40,412)	6.5 (5.87 to 7.28)	116,546 (105,505 to 128,325)	9.08 (8.22 to 9.94)	1.15 (1.11 to 1.19)
Middle SDI	83,375 (76,841 to 90,067)	8.77 (8.1 to 9.44)	279,778 (251,149 to 306,137)	11.98 (10.76 to 13.09)	1.24 (1.1 to 1.37)
High-middle SDI	162,685 (155,451 to 168,785)	16.08 (15.28 to 16.7)	326,640 (299,662 to 349,530)	16.24 (14.9 to 17.39)	−0.04 (−0.15 to 0.06)
High SDI	222,296 (211,298 to 228,215)	21.18 (20.08 to 21.75)	327,570 (294,904 to 345,578)	16.29 (14.93 to 17.09)	−1.09 (−1.17 to −1)
Region					
Andean Latin America	1523 (1345 to 1690)	7.88 (6.97 to 8.77)	5630 (4593 to 6791)	10.32 (8.41 to 12.43)	1.16 (1.02 to 1.3)
Australasia	5639 (5377 to 5814)	24.41 (23.19 to 25.2)	8382 (7575 to 8978)	16.24 (14.82 to 17.29)	−1.73 (−1.88 to −1.59)
Caribbean	3286 (3113 to 3415)	13.13 (12.36 to 13.65)	7995 (6935 to 9176)	15.46 (13.41 to 17.75)	0.63 (0.57 to 0.68)
Central Asia	5122 (4952 to 5301)	10.96 (10.57 to 11.34)	7467 (6822 to 8166)	11.21 (10.27 to 12.22)	0.33 (0.17 to 0.5)
Central Europe	30,827 (29,768 to 31,602)	21.53 (20.69 to 22.11)	51,567 (45,636 to 57,749)	23.57 (20.8 to 26.42)	0.32 (0.21 to 0.42)
Central Latin America	5732 (5482 to 5890)	7.3 (6.9 to 7.54)	22,470 (19,542 to 25,997)	9.71 (8.42 to 11.22)	0.97 (0.93 to 1.01)
Central Sub-Saharan Africa	1515 (1209 to 1898)	7.55 (6.05 to 9.42)	3544 (2705 to 4609)	7.45 (5.74 to 9.87)	−0.11 (−0.36 to 0.14)
East Asia	83,307 (73,752 to 93,316)	10.28 (9.16 to 11.43)	275,604 (238,238 to 317,886)	14.1 (12.24 to 16.15)	1.4 (1.17 to 1.64)
Eastern Europe	49,828 (48,379 to 51,349)	18.01 (17.45 to 18.57)	63,476 (57,180 to 70,011)	18.3 (16.47 to 20.2)	−0.31 (−0.51 to −0.12)
Eastern Sub-Saharan Africa	4900 (4103 to 5787)	7.04 (5.87 to 8.24)	12,717 (10,940 to 15,001)	8.47 (7.38 to 9.9)	0.69 (0.62 to 0.76)
High-income Asia Pacific	34,338 (32,599 to 35,180)	17.94 (16.9 to 18.45)	76,929 (64,821 to 83,603)	15.29 (13.42 to 16.36)	−0.68 (−0.75 to −0.62)
High-income North America	71,908 (67,815 to 74,170)	20.01 (18.91 to 20.61)	95,664 (88,321 to 99,688)	14.94 (13.94 to 15.51)	−1.22 (−1.32 to −1.13)
North Africa and Middle East	13,079 (11,006 to 15,347)	8.2 (6.87 to 9.56)	39,147 (34,761 to 44,107)	9.83 (8.68 to 11.05)	0.81 (0.62 to 1)
Oceania	206 (161 to 248)	7.74 (6.1 to 9.24)	551 (443 to 682)	8.81 (7.19 to 10.65)	0.41 (0.34 to 0.49)
South Asia	27,309 (24,144 to 31,138)	5.34 (4.66 to 6.1)	94,846 (81,524 to 109,075)	7.29 (6.25 to 8.33)	0.91 (0.79 to 1.04)
Southeast Asia	23,639 (20,959 to 25,972)	9.82 (8.72 to 10.73)	82,024 (67,617 to 94,606)	14.4 (11.89 to 16.62)	1.25 (1.19 to 1.31)
Southern Latin America	8829 (8469 to 9093)	19.95 (18.99 to 20.6)	17,930 (16,774 to 18,975)	21.22 (19.87 to 22.44)	0.15 (0.05 to 0.25)
Southern Sub-Saharan Africa	2569 (2213 to 3044)	10.17 (8.66 to 12.21)	5922 (5329 to 6580)	11.52 (10.4 to 12.74)	0.45 (0.24 to 0.66)
Tropical Latin America	8475 (8117 to 8749)	10.1 (9.57 to 10.47)	27,704 (25,668 to 29,090)	11.68 (10.78 to 12.28)	0.6 (0.44 to 0.75)
Western Europe	130,906 (124,426 to 134,404)	22.24 (21.14 to 22.83)	172,454 (155,345 to 181,815)	17.32 (15.83 to 18.13)	−1.09 (−1.25 to −0.92)
Western Sub-Saharan Africa	5189 (4250 to 6327)	6.61 (5.44 to 7.98)	13,773 (11,698 to 16,069)	8.41 (7.27 to 9.67)	1.05 (0.95 to 1.15)

Abbreviations: SDI, sociodemographic index; ASDR, age-standardized death rate; EAPC, estimated annual percentage change; UI, uncertainty interval; CI, confidence interval.

**Table 2 nutrients-16-00132-t002:** The temporal trends of EAPC of ARDS attributed to dietary risks across different SDI regions.

	All Dietary Risks (95% CI)	Diet Low in Whole Grains (95% CI)	Diet Low in Milk (95% CI)	Diet Low in Fiber (95% CI)	Diet Low in Calcium (95% CI)	Diet High in Red Meat (95% CI)	Diet High in Processed Meat (95% CI)
Global	−0.29 (−0.36 to −0.22)	−0.29 (−0.36 to −0.23)	0.19 (0.16 to 0.23)	−1.13 (−1.23 to −1.02)	0.04 (−0.06 to 0.13)	−0.32 (−0.37 to −0.28)	−1.08 (−1.2 to −0.96)
Low SDI	0.44 (0.4 to 0.48)	0.56 (0.51 to 0.6)	0.46 (0.38 to 0.54)	0.12 (−0.03 to 0.27)	0.3 (0.25 to 0.34)	0.81 (0.72 to 0.9)	2.47 (2.32 to 2.62)
Low-middle SDI	0.89 (0.84 to 0.94)	1.08 (1.04 to 1.12)	1.13 (1.08 to 1.18)	−0.28 (−0.42 to −0.15)	0.65 (0.58 to 0.71)	2.38 (2.28 to 2.48)	0.89 (0.84 to 0.94)
Middle SDI	0.96 (0.82 to 1.11)	1.24 (1.1 to 1.39)	1.13 (1 to 1.25)	−0.61 (−0.75 to −0.46)	0.51 (0.35 to 0.66)	2.95 (2.77 to 3.12)	1.79 (1.71 to 1.86)
High-middle SDI	−0.32 (−0.43 to −0.2)	−0.34 (−0.43 to −0.25)	0.55 (0.49 to 0.61)	−0.92 (−1.16 to −0.69)	−0.37 (−0.54 to −0.21)	0.09 (0.01 to 0.17)	−1.04 (−1.19 to −0.89)
High SDI	−1.04 (−1.11 to −0.98)	−0.97 (−1.05 to −0.89)	−1.14 (−1.2 to −1.09)	−1.5 (−1.54 to −1.47)	−0.85 (−0.89 to −0.82)	−1.29 (−1.37 to −1.21)	−0.88 (−1.01 to −0.74)

Abbreviations: SDI, sociodemographic index; ASDR, age-standardized death rate; EAPC, estimated annual percentage change; UI, uncertainty interval; CI, confidence interval.

## Data Availability

The datasets generated and/or analyzed during the current study are available in the Global Health Data Exchange query tool, http://ghdx.healthdata.org/gbd-results-tool, accessed on 22 January 2022.

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
