# Peer review of "Distributions and Trends of the Global Burden of Colorectal Cancer Attributable to Dietary Risk Factors over the Past 30 Years"

_nutrients, 2023, doi:10.3390/nu16010132_

Round 1

Reviewer 1 Report

Comments and Suggestions for Authors

Major critique

It is not clear how your observations are different from what was already reported by the GBD in Lancet Gastroenterol Hepatol. 2022 Jul;7(7):627-647, titled “Global, regional, and national burden of colorectal cancer and its risk factors, 1990-2019: a systematic analysis for the Global Burden of Disease Study 2019”

Minor critiques

The following sentence in the Introduction is confusing as it says both increased and decreased CRC incidence in high SDI “In general, the number of CRC-related incident cases and death cases increased in all socio-demographic indices (SDI) regions. The incidence and mortality rates mostly declined in high SDI countries, while age-standardized rates increased in some low SDI and low-middle SDI countries and regions [2–4].”

Consider replacing “putting” with a more appropriate word in the Introduction “and putting dietary advice and medical decisions on the basis of epidemiological studies is essential.”

Correct date to data in “2.2 The GBD dietary date were based on the FAO Food Balance Sheet (FBS) and Global Nutrient Database.”

Correct the sentence in 2.4 “Patient and public involvement Global Burden of Disease (GBD) Study is an international scientific collaboration.” Not sure why you need this section if patients were not involved in your study.

Some of panels in Figure 3 are low resolution and need to be improved.

In Discussion, you talk about the increase in CRC deaths in low SDI regions while there is a decrease in high SDI regions but don't delve into the potential reasons or implications. It's also essential to conclude by suggesting directions for future research.

Comments on the Quality of English Language

Can be improved

Author Response

Dear Reviewer:

We would like to thank for you work devoted to our manuscript entitled "Distributions and trends of the global burden of colorectal cancer attributable to dietary risk factors over the past 30 years" (nutrients-2659614). Those comments are all valuable and very helpful for revising and improving our paper. We have studied the comments carefully and made corrections that we hope to meet with approval. All the revised portions are highlighted in the paper.

Each concern is discussed in detail below. We want to highlight that significant revisions have been made to the discussion section, addressing and expanding upon key aspects highlighted in our study. Moreover, we have enlisted the assistance of English language editing services to enhance the overall linguistic quality of the manuscript. This step has been taken to ensure that the manuscript adheres to the highest publication standards and is more accessible to a wider audience. These revisions were performed to address the reviewers’ concerns. Thank you again for allowing us to resubmit our manuscript for your consideration.

Detailed Responses to Reviewer

Major critique

It is not clear how your observations are different from what was already reported by the GBD in Lancet Gastroenterol Hepatol. 2022 Jul;7(7):627-647, titled “Global, regional, and national burden of colorectal cancer and its risk factors, 1990-2019: a systematic analysis for the Global Burden of Disease Study 2019”

Response: Thank you for your insightful comments and for referencing the study titled "Global, regional, and national burden of colorectal cancer and its risk factors, 1990-2019: a systematic analysis for the Global Burden of Disease Study 2019" published in Lancet Gastroenterol Hepatol [1].

Our study, "Distributions and trends of the global burden of colorectal cancer attributable to dietary risk factors over the past 30 years," aims to offer a focused examination on the specific influence of dietary factors on the global burden of colorectal cancer. By honing in on the role of dietary factors in the incidence and trends of colorectal cancer globally, our study delves into the nuanced relationships between various dietary elements and the prevalence of this malignancy. We believe that our research provides additional granularity by emphasizing the specific contributions and variations of dietary risk factors to the burden of colorectal cancer, complementing the broader landscape outlined in the GBD study.

Appropriately, we referenced this piece of literature and highlighted the differences between the two articles in Discussion. We discuss in detail the implications of our findings regarding the epidemiology of colorectal cancer (CRC) associated with dietary factors across diverse regions, gender, and age groups on a global scale. Our aim is to underscore the relevance of these findings in shaping more targeted and efficacious dietary interventions for CRC prevention. Your comprehensive review and insightful feedback are greatly appreciated.

Minor critiques

The following sentence in the Introduction is confusing as it says both increased and decreased CRC incidence in high SDI “In general, the number of CRC-related incident cases and death cases increased in all socio-demographic indices (SDI) regions. The incidence and mortality rates mostly declined in high SDI countries, while age-standardized rates increased in some low SDI and low-middle SDI countries and regions [2–4].”

Response: Thank you very much for pointing out our problem. We are very sorry that our previous expression may not be appropriate and may cause readers to misunderstand. Indeed, the implication is that the mortality and incidence of colorectal cancer are increasing globally. But in countries with high SDI, the incidence and mortality rates are decreasing, in contrast to the general trend, while age-standardized rates increased in some low SDI and low-middle SDI countries and regions. We have revised and highlighted our presentation in the Introduction. As follows:

However, a more precise and comprehensive comprehension of the dietary risks linked to CRC remains incomplete, underscoring the indispensable nature of relying on epidemiological studies to shape dietary recommendations and guide medical decision-making.

Consider replacing “putting” with a more appropriate word in the Introduction “and putting dietary advice and medical decisions on the basis of epidemiological studies is essential.”

Response: Thank you for your suggestion regarding the choice of wording in the Introduction section, specifically the phrase "putting dietary advice and medical decisions on the basis of epidemiological studies is essential."

In our revised manuscript, we will carefully consider and replace with a more precise and fitting sentence that better conveys the importance of basing dietary advice and medical decisions on epidemiological studies. We appreciate your attention to detail and strive to enhance the clarity and accuracy of our manuscript. As follows:

However, a more precise and comprehensive comprehension of the dietary risks linked to CRC remains incomplete, underscoring the indispensable nature of relying on epidemiological studies to shape dietary recommendations and guide medical decision-making”

Correct date to data in “2.2 The GBD dietary date were based on the FAO Food Balance Sheet (FBS) and Global Nutrient Database.”

Response: Thank you for your thorough review of our manuscript. We have duly noted the spelling error you pointed out and have corrected it in the revised version. We appreciate your attention to detail, and we are committed to ensuring a more accurate and clear manuscript in the final version.

Correct the sentence in 2.4 “Patient and public involvement Global Burden of Disease (GBD) Study is an international scientific collaboration.” Not sure why you need this section if patients were not involved in your study.

Response: Thank you for your valuable feedback. We agree with your opinion that this paragraph is slightly redundant here, and we have deleted it in the manuscript We believe this adjustment enhances the focus of our study. We appreciate your constructive input and are grateful for your continued support in improving our manuscript. If you have any further comments or questions, please feel free to share them.

Some of panels in Figure 3 are low resolution and need to be improved.

Response: Thank ·you for your meticulous review and for highlighting the resolution concerns regarding specific panels in Figure 3. We sincerely apologize for the low resolution in these panels. We understand the critical importance of visual clarity in our figures and have taken immediate action to rectify this issue. All images throughout the revised manuscript have been thoroughly reviewed, ensuring that high-resolution versions of every image, including those in Figure 3, have been uploaded. Your attention to detail is immensely valuable, and your feedback is pivotal in elevating the quality of our work.

In Discussion, you talk about the increase in CRC deaths in low SDI regions while there is a decrease in high SDI regions but don't delve into the potential reasons or implications. It's also essential to conclude by suggesting directions for future research.

Response: Thank you for your insightful feedback on our Discussion section. We acknowledge the need to delve deeper into the potential reasons behind the increase in colorectal cancer (CRC) deaths in low Socio-demographic Index (SDI) regions juxtaposed with the decrease in high SDI regions. In the Discussion section of our revised manuscript, we will thoroughly explore plausible explanations for these divergent trends. This will include analyzing various factors such as healthcare access, socioeconomic disparities, lifestyle differences, and healthcare infrastructure that might contribute to this phenomenon [2]. Additionally, we will underscore the implications of these trends, emphasizing their significance in shaping regional healthcare policies and interventions aimed at reducing CRC mortality rates globally. We appreciate your valuable insights, and we are dedicated to incorporating these enhancements into our manuscript revisions.

Comments on the Quality of English Language:Can be improved.

Response: Thank you for your valuable and insightful feedback. In the revised manuscript, we have collaborated with a native English-speaking colleague who meticulously reviewed and enhanced the language throughout the document.

We appreciate for Editors/Reviewers’ warm work earnestly and hope that the correction will meet with approval.

Once again, thank you very much for your comments and suggestion.

Yours Sincerely,

Fuling Zhou, MD., Ph.D.

Yongchang Wei, MD., Ph.D.

Wuhan University

Reviewer 2 Report

Comments and Suggestions for Authors

In this study, the authors conducted an analysis of epidemiological data related to colorectal cancer (CRC), taking into account variables such as gender, region, country, age, and dietary risks. The data were sourced from the Global Burden of Disease (GBD). It is imperative to gain a comprehensive understanding of the disease burden of CRC attributed to dietary risks. This knowledge is essential for governments and health organizations to develop effective CRC prevention strategies and guide individuals toward adopting healthy dietary habits.

Regarding specific comments, the manuscript acknowledges gender differences in CRC mortality but lacks a thorough discussion of the underlying reasons. A more comprehensive exploration is necessary to elaborate on potential genetic and environmental factors contributing to these differences.

Additionally, the manuscript observes a decrease in the global age-standardized death rate (ASDR) of CRC attributable to dietary risks, alongside an increase in CRC deaths, especially among young adults. The manuscript should provide possible explanations for this phenomenon and discuss its implications. Considering factors such as evolving dietary patterns, lifestyle choices, and screening practices could offer insights into the observed trends. Understanding these dynamics is crucial for developing targeted interventions to address the growing CRC burden, particularly among young adults.

Author Response

Dear Editors and Reviewers:

We would like to thank your work devoted to our manuscript entitled "Distributions and trends of the global burden of colorectal cancer attributable to dietary risk factors over the past 30 years" (nutrients-2659614). Those comments are all valuable and very helpful for revising and improving our paper. We have studied the comments carefully and made corrections that we hope to meet with approval. All the revised portions are highlighted in the paper.

Each concern is discussed in detail below. We want to highlight that significant revisions have been made to the discussion section, addressing and expanding upon key aspects highlighted in our study. Moreover, we have enlisted the assistance of English language editing services to enhance the overall linguistic quality of the manuscript. This step has been taken to ensure that the manuscript adheres to the highest publication standards and is more accessible to a wider audience. These revisions were performed to address the reviewers’ concerns. Thank you again for allowing us to resubmit our manuscript for your consideration.

In this study, the authors conducted an analysis of epidemiological data related to colorectal cancer (CRC), taking into account variables such as gender, region, country, age, and dietary risks. The data were sourced from the Global Burden of Disease (GBD). It is imperative to gain a comprehensive understanding of the disease burden of CRC attributed to dietary risks. This knowledge is essential for governments and health organizations to develop effective CRC prevention strategies and guide individuals toward adopting healthy dietary habits.

Response: Thank you for your thoughtful assessment of our study. We wholeheartedly agree on the significance of our analysis. We appreciate your recognition of the critical significance of our study's contribution to advancing strategies for CRC prevention and look forward to incorporating your valuable feedback into our revisions.

Regarding specific comments, the manuscript acknowledges gender differences in CRC mortality but lacks a thorough discussion of the underlying reasons. A more comprehensive exploration is necessary to elaborate on potential genetic and environmental factors contributing to these differences.

Response: Thank you for your insightful comments on our manuscript. We acknowledge the importance of addressing gender differences in colorectal cancer (CRC) mortality comprehensively. In our revised manuscript, the Discussion section will undergo significant modifications to delve deeper into the underlying reasons behind these differences. We will provide a more extensive exploration of potential genetic and environmental factors contributing to this phenomenon, including hormonal variances, lifestyle factors, and their potential interaction with genetic predispositions, offering a more nuanced understanding of the observed gender disparities in CRC mortality rates. We appreciate your guidance in enhancing the depth of our discussion on this critical aspect of CRC research.

Additionally, the manuscript observes a decrease in the global age-standardized death rate (ASDR) of CRC attributable to dietary risks, alongside an increase in CRC deaths, especially among young adults. The manuscript should provide possible explanations for this phenomenon and discuss its implications. Considering factors such as evolving dietary patterns, lifestyle choices, and screening practices could offer insights into the observed trends. Understanding these dynamics is crucial for developing targeted interventions to address the growing CRC burden, particularly among young adults.

Response: Thank you for highlighting the contrasting trends observed in our manuscript regarding the decrease in the global age-standardized death rate (ASDR) of colorectal cancer (CRC) linked to dietary risks, alongside an increase in CRC deaths among young adults. In the Discussion of our revised manuscript, we will succinctly explore possible reasons for this phenomenon, considering evolving dietary patterns, lifestyle choices, and screening practices as contributing factors [3,4]. Understanding these dynamics is crucial for developing targeted interventions to address the growing burden of CRC, especially among young adults. We appreciate your valuable insights and will ensure a more concise yet comprehensive discussion in our manuscript revisions as follows:

The rise in early-onset colorectal cancer mortality could be attributed to diverse factors including unhealthy lifestyle choices, genetic predispositions, and environmental exposures. Specific dietary habits, such as increased intake of red and processed meats along with inadequate fiber consumption, may contribute to this trend. Moreover, insufficient screening and delayed diagnosis in younger individuals might exacerbate mortality rates related to early-onset colorectal cancer. Obesity, and lack of exercise were probable major contributors to CRC in young adults.

We appreciate for Reviewers’ warm work earnestly and hope that the correction will meet with approval.

Once again, thank you very much for your comments and suggestion.

Yours Sincerely,

Fuling Zhou, MD., Ph.D.

Yongchang Wei, MD., Ph.D.

Wuhan University

Reviewer 3 Report

Comments and Suggestions for Authors

In this manuscript the authors present estimates of the association between diet and cancer developed with data from the Global Burden of Disease.

In the presentation, enriched by very substantial additional material, the authors illustrate the sources of the data used quite comprehensively - i.e. mortality data, sociodemographic indicators, eating habits - and indicate the statistical analysis methods.

However, I believe it could be useful for the authors to explain in more detail how the characteristics of the diet are used, in particular illustrating the sources of food data and how these have been synthesized to identify diets with different contents of cereals, milk, fibre, calcium , red meats, and processed meats. It may be sufficient to provide more direct bibliographical indications than that presented as reference 23

Author Response

Dear Editors and Reviewers:

We would like to thank reviewer’ work devoted to our manuscript entitled "Distributions and trends of the global burden of colorectal cancer attributable to dietary risk factors over the past 30 years" (nutrients-2659614). Those comments are all valuable and very helpful for revising and improving our paper. We have studied the comments carefully and made corrections that we hope to meet with approval. All the revised portions are highlighted in the paper.

Each concern is discussed in detail below. We want to highlight that significant revisions have been made to the discussion section, addressing and expanding upon key aspects highlighted in our study. Moreover, we have enlisted the assistance of English language editing services to enhance the overall linguistic quality of the manuscript. This step has been taken to ensure that the manuscript adheres to the highest publication standards and is more accessible to a wider audience. These revisions were performed to address the reviewers’ concerns. Thank you again for allowing us to resubmit our manuscript for your consideration.

In this manuscript the authors present estimates of the association between diet and cancer developed with data from the Global Burden of Disease.

Response: Thank you for highlighting the core focus of our manuscript, which centers on presenting estimates of the association between diet and CRC, derived from the Global Burden of Disease data. We appreciate your attention to this fundamental aspect of our research.

In the presentation, enriched by very substantial additional material, the authors illustrate the sources of the data used quite comprehensively - i.e. mortality data, sociodemographic indicators, eating habits - and indicate the statistical analysis methods.

Response: It's encouraging to receive your positive feedback regarding the thorough presentation of data sources and the delineation of statistical analysis methods. We are glad to know it has been beneficial. Thank you for acknowledging these efforts in our manuscript.

However, I believe it could be useful for the authors to explain in more detail how the characteristics of the diet are used, in particular illustrating the sources of food data and how these have been synthesized to identify diets with different contents of cereals, milk, fibre, calcium, red meats, and processed meats. It may be sufficient to provide more direct bibliographical indications than that presented as reference 23

Response: Thank you for your insightful feedback and suggestions. We acknowledge the need to explain in more detail how dietary characteristics are exploited, particularly in describing the sources of food data and their synthesis to identify a variety of diets containing cereals, milk, fiber, calcium, red meat, and processed meat.

In response to your suggestion, we will extend the methodology section to more fully describe the sources of the food data used in our study. In addition, we present the exposure values for each food risk factor in the Supplementary material Table S1. We appreciate your valuable input and will ensure that these enhancements are incorporated into our manuscript to improve the clarity and depth of our approach.

Supplementary Table S1. The definition and theoretical minimum risk exposure level for dietary risk factor in 2019.

Dietary risks

Definition

Theoretical minimum risk exposure level

Diet low in whole grains

Average daily consumption of whole grains (bran, germ, and endosperm in their natural proportion) from breakfast cereals, bread, rice, pasta, biscuits, muffins, tortillas, pancakes, and other sources

Consumption of whole grains 100-150 g per day

Diet low in milk

Average daily consumption of milk, including non-fat, low-fat, and full-fat milk, excluding soy milk and other plant derivatives

Consumption of milk 350-520 g per day

Diet low in fiber

Average daily intake of fibre from all sources including fruits, vegetables, grains, legumes, and pulses

Consumption of fibre 19-28 g per day

Diet low in calcium

Average daily intake of calcium from all sources, including milk, yogurt, and cheese

Consumption of calcium 1.0-1.5 g per day

Diet high in red meat

Average daily consumption of red meat (beef, pork, lamb, and goat but excluding poultry, fish, eggs, and all processed meats)

Consumption of red meat 18-27 g per day

Diet high in processed meat

Average daily consumption of meat preserved by smoking, curing, salting, or addition of chemical preservatives

Consumption of processed meat 0-4 g per day

We appreciate for Reviewers’ warm work earnestly and hope that the correction will meet with approval.

Once again, thank you very much for your comments and suggestion.

Yours Sincerely,

Fuling Zhou, MD., Ph.D.

Yongchang Wei, MD., Ph.D.

Wuhan University

Reviewer 4 Report

Comments and Suggestions for Authors

This nice retrospective study examines a link between dietary habits and colorectal cancer. The figures and tables are very relevant. I have some comments:

Methods: Is this the correct equation to calculate ASDR?

Results: The world maps and their legends are too small to read clearly. A schematic representation could provide a solution.

Discussion: The results of the Global Burden of Disease Study, as conducted by the IHME and already used in quite some retrospective trials, are not entirely undisputed by the WHO, and this should be mentioned in the Discussion.

Comments on the Quality of English Language

Some grammatical errors throughout the article.

Round 2

Reviewer 1 Report

Comments and Suggestions for Authors

The revised manuscript is much improved.

Comments on the Quality of English Language

Additional minor edits to language and grammar is recommended.

Author Response

Dear Reviewer,

Thank you for your valuable feedback. We appreciate your thorough review and have carefully considered your suggestions. We had made additional minor edits to further refine the language and grammar in the manuscript. Your insights have been immensely helpful in improving the quality of our work.

Best regards,

Fuling Zhou; Yongchang Wei